# Butyrate modulates mucin secretion and bacterial adherence in LoVo cells via MAPK signaling

**Tae-Hwan Jung[1], Kyoung-Sik Han[1], Jeong-Hyeon Park[2], Hyo-Jeong Hwang[1] ***

**1** Department of Food and Nutrition, Sahmyook University, Seoul, Korea, **2** Department of Biological Sciences, Xi'an Jiaotong-Liverpool University, Suzhou, China

* hjhwang@syu.ac.kr

**Data Availability Statement:** All relevant data are within the paper.

**Funding:** This paper was supported by the Academic Research Fund of Dr. Myung Ki (MIKE) Hong in 2020. The funders had no role in study

## Abstract

Short-chain fatty acids contribute to normal bowel function and prevent bacterial infections. In particular, butyrate is a promising candidate that plays an important role in regulating the functional integrity of the gastrointestinal tract by stimulating mucin secretion. We investigated whether butyrate treatment modulates mucin secretion and bacterial adherence in LoVo cells. In addition, the possible signaling pathways were also examined in connection with the upregulation of mucin secretion. The results showed that butyrate induced mucin secretion in LoVo cells, resulting in the inhibition of *Escherichia coli* adhesion by increasing the adherence of *Lactobacillus acidophilus* and *Bifidobacterium longum*. The gene expression analysis suggests that mitogen-activated protein kinase (MAPK) signaling pathways including Cdc42-PAK pathway appears to be involved in stimulating mucin secretion. More importantly, butyrate induced the increased actin expression and polymerization in LoVo cells, which could be attributable to the Cdc42-PAK signaling pathway, implicated in actin cytoskeleton and mucin secretion. Our results provide a molecular basis in modulating bacterial adherence and the MAPK signaling pathway for the improved homeostasis of colonic epithelial cells.

## Introduction

Short-chain fatty acids (SCFAs) are the primary metabolites of fermentation of non-digestible carbohydrates and mainly consist of acetic acid, propionic acid, and butyric acid. These SCFAs contribute to the maintenance of bowel function and help prevent infections [1]. In particular, butyrate is known to be a major regulator of homeostasis in the gut and can exert actions related to cellular homeostasis, anti-inflammatory, and anti-carcinogenic functions [2, 3]. Colonocytes utilize butyrate as their major energy source, and most butyrate in the gut is metabolized by the colonic epithelium. It has also been reported that butyrate plays an important role in cellular responses through the activation of mitogen-activated protein kinase (MAPK) signaling pathways [4, 5].

The composition of the gut microbiota plays an important role in the maintenance of gut homeostasis [6]. It has been reported that the gut microbial community is associated with

design, data collection and analysis, decision to publish, or preparation of the manuscript.

**Competing interests:** The authors have declared that no competing interests exist.

diseases including necrotizing enterocolitis and inflammatory bowel disease [7]. The predominance of beneficial bacteria can positively impact the overall health of the gut. Butyrate contributes to the prevention and improvement of intestinal disease, and promotes the growth of beneficial bacteria, including *Lactobacillus* and *Bifidobacterium* strains in the gut [8]. However, for these bacterial strains to grow, they need to adhere and survive in the gastrointestinal (GI) tract. The intestinal mucus layer could be a primary factor in successful adhesion of the gut microbiota, and it has been reported that butyrate can stimulate mucus production in the GI tract [9, 10].

Mucins, the major constituent of the mucus layer, are large, highly glycosylated proteins with over 80% of their mass with carbohydrates [11]. They are concentrated in the mucus layer [12] and play an important role in luminal protection in the GI tract where they are considered a primary protector against pathogens and harmful substances [13]. The mucus layer in the colon consists of two layers. The less-dense outer layer is a habit of the commensal bacteria where other bacteria do not adhere well [14]. An inner mucus layer in the colon is firmly attached to the epithelial cells where it is impermeable to bacteria as a major barrier against pathogens [15]. These properties of the mucus layer play an important role in protecting intestinal homeostasis and confer the antibacterial barrier [16]. A variety of studies have reported that mucin secretion is important in epithelial cell physiology in the GI tract [17]; however, the molecular mechanism by which how butyrate affects intestinal mucin secretion is not well known. Therefore, we investigated the effect of butyrate on the attachment of probiotics and their inhibitory effect against pathogens in LoVo human colorectal cells. In addition, the signaling pathways involved in butyrate-dependent mucin secretion were analyzed using real-time polymerase chain reaction (PCR) and proteomic analysis.

## Materials and methods

### Cell culture

The LoVo human colorectal cell line was obtained from the Korean Cell Line Bank (Seoul, Korea). All media and reagents were purchased from Thermo Fisher Scientific (Waltham, MA, USA). Cells were grown in Roswell Park Memorial Institute (RPMI) 1640 media supplemented with 10% heat-inactivated fetal bovine serum, 1% penicillin (100 U/mL), and streptomycin (100 U/mL) at 37˚C with 5% $CO_2$ in a humidified atmosphere. For immunohistochemical staining, measurement of mucin2, and bacterial adherence assays, LoVo cells ($1 \times 10^6$ cells/mL) were cultured in a 6-well plate (Thermo Fisher Scientific, USA) for 3 days, followed by the addition of butyrate in RPMI 1640 medium at final concentrations of 0, 2, 4, 6, or 8 mM and then cultured for 48 h. As a control, RPMI 1640 medium was added to a 6-well plate in place of butyrate.

### Bacterial adherence assays

Bacteria were purchased from the Korean Culture Center of Microorganisms (Seoul, Korea). *Escherichia coli* ATCC 43896 was cultured in tryptic soy broth (BD Difco, Bergen County, NJ, USA) under aerobic conditions for 18 h at 37˚C. *Lactobacillus acidophilus* ATCC 4356 and *Bifidobacterium longum* ATCC 15707 were cultured in Lactobacilli MRS broth (BD Difco) and Reinforced Clostridial Medium (RCM), (Oxoid, Basingstoke, Hampshire, UK) under anaerobic conditions for 18 h at 37˚C. For the bacterial adherence assay, all bacteria were subcultured three times. *L. acidophilus* or *B. longum* ($1 \times 10^8$ cfu/well) were added to butyrate-treated LoVo cells in 6-well plates, followed by incubation for 3 h at 37˚C. Each well was washed three times with phosphate-buffered saline (PBS) to remove the non-bound bacteria. Adherent *L. acidophilus* or *B. longum* were counted on Lactobacilli MRS or RCM agar using the pour plate

method. To analyze the inhibitory effect of *L. acidophilus* or *B. longum* on the adherence of *E. coli* to LoVo cells, *L. acidophilus* or *B. longum* were incubated with butyrate-treated LoVo cells for 3 h at 37˚C, each well was washed three times with PBS, followed by the addition of *E. coli* ($1 \times 10^8$ cfu/well) and then incubated for 3 h at 37˚C. To remove non-bound *E. coli*, each well was washed three times with PBS. Adherent *E. coli* were counted on MacConkey agar (BD Difco) using the pour plate method.

## Measurement of mucin

We measured the mucin2 that secreted outside the LoVo cells to investigate stimulated by the butyrate treatment in the LoVo cells. The Control and butyrate-treated LoVo cells culture fluid were collected in microtube and then centrifuged at 10,000 rpm for 10 min. Mucin2 content in the supernatant was analyzed using an enzyme-linked immunosorbent assay (ELISA) kit for human MUC2 (USCN Life Science Inc., Wuhan, China) according to the manufacturer's instructions, and the kit is a sandwich enzyme immunoassay for *in vitro* quantitative measurement of mucin2. Briefly, appropriately diluted standards and sample solutions (100 μL) were added to a ready-to-use 96-well strip plate pre-coated to capture antibody for mucin2, and then incubated for 1 h at 37˚C. The liquid was removed from each well and 100 μL of biotin-conjugated detection antibody solution specific to mucin2 was added, then incubated for 30 min at 37˚C. Each well was washed three times with wash buffer and 100 μL of Avidin-Horseradish Peroxidase (HRP) conjugate solution was added to each well and incubated for 30 min at 37˚C. The wells were washed five times with wash buffer and 90 μL TMB substrate solution was added to each well and incubated for 15 min at 37˚C, followed by the addition of 50 μL stop solution. The absorbance was read at 450 nm using a microplate reader (Molecular Devices, San Jose, CA, USA).

## Immunohistochemical staining

Immunohistochemical staining of LoVo cells was performed using a VECTASTAIN Elite ABC kit (Vector Laboratories, Burlingame, CA, USA) according to the manufacturer's guidelines. Briefly, cells were incubated for 20 min with blocking serum, then excess serum was removed. Cells were incubated with primary antibody specific to mucin2 for 30 min then rinsed with wash buffer for 5 min. A biotinylated secondary antibody specific to mucin2 was added to each well and incubated for 30 min, followed by rinsing with wash buffer for 5 min. Cells were incubated in VECTASTAIN Elite ABC reagent for 30 min and washed for 5 min. Stained cells were observed under a microscope (Olympus BX51, Tokyo, Japan).

## Investigation of mucin secretion pathway

Total RNA was extracted from cells using TRIzol reagent (Sigma, St. Louis, MO, USA) and a PureLink RNA mini kit (Life Technologies, Carlsbad, CA, USA) according to the manufacturer's guidelines. The first strand of cDNA was synthesized using a high-capacity RNA-to-cDNA kit (Applied Biosystems, Foster City, CA, USA). 2X RT buffer mix (10 μL), 20X RT enzyme mix (1 μL), RNA (1–9 uL), and nuclease-free water were added to yield a final volume of 20 μL. RT was performed at 37˚C for 60 min and at 95˚C for 5 min using a Veriti 97 Well Thermal Cycler (Applied Biosystems). The pathway of mucin genes induced by butyrate treatment was investigated using a TaqMan® array 96-well fast plate (Applied Biosystems) and the StepOnePlus™ Real-Time PCR System (Applied Biosystems). Each well containing 4 μL of Taqman master mix, 1 μL of diluted cDNA, and 1 μL of DNase-free water was preincubated at 50˚C for 2 min and at 95˚C for 10 min, followed by 40 cycles of 95˚C for 15 s and 60˚C for 1 min. The mucin secretion pathway was analyzed using the $2^{-\text{delta delta Ct}}$ method. The delta Ct

value was estimated by subtracting the GAPDH Ct value from the Ct value of the target gene. The delta delta Ct value was calculated by subtracting the delta Ct of the control sample from that of the butyrate-treated sample. The fold change of the butyrate-treated sample relative to the control sample was calculated by the $2^{-\text{delta delda Ct}}$ method. The decision about the significant changes in mucin secretion was determined if an over 2-fold increase was observed compared to the control sample.

## Proteomics

Two-dimensional gel electrophoresis was used to study the properties of the proteins. For analysis in the first dimension, protein samples from control and butyrate-treated cells were electrofocused on immobilized pH gradient strips (pH 3–10). Then in the second dimension, isoelectric focusing strips were electrophoresed on 9% –16% gradient polyacrylamide gels until the dye migrated to the lower end of the gel. The relative abundance of protein spots was quantified by staining with Coomassie Blue G-250, then scanned with a GS-710 imaging calibrated densitometer (Bio-Rad, Hercules, CA, USA) and analyzed with ImageMaster 2D platinum software (Amersham Biosciences Amersham, Buckingham, UK). The target protein spot on the gel was excised, destained using 25 mM ammonium bicarbonate containing 50% acetonitrile, and then digested with 0.1-fold trypsin per protein in 50 mM ammonium bicarbonate buffer. The protein solution was evaporated, and the remaining peptide was dissolved in 3% formic acid. Protein identification was performed using liquid chromatography (LC) and mass spectrometry (MS) with an LTQ-Orbitrap XL MS (Thermo Fisher Scientific) connected to an EASY-nLC 1000 system (Thermo Fisher Scientific). A C18 silica-packed column (ZORBAX 300SB-C18, 150 mm × 0.1 mm, 3- μm pore size) was obtained from Agilent Technologies (Santa Clara, CA, USA). Mobile phase A for LC separation consisted of 0.1% formic acid and 3% acetonitrile in deionized water, and mobile phase B was 0.1% formic acid in acetonitrile. The chromatography gradient was programmed to increase linearly from 0% B to 32% B in 40 min, 32% B to 60% B in 4 min, 60% B to 95% B in 4 min, 95% B in 4 min, and 0% B in 6 min. The flow rate was maintained at 1,500 nL/min. MS data were processed using Proteome Discover (Thermo Fisher Scientific). Mascot software and the NCBI database were used to identify the proteins.

## Statistical analysis

The results are presented as mean ± SEM. Any statistically significant differences between the groups was determined using the SAS/PROC GLM software (SAS version 9.1; SAS Institute Inc., Cary, NC, USA). The statistical significance of bacterial adherence and mucin protein was analyzed by one-way ANOVA with Duncan's multiple range test.

## Results

### Bacterial adherence

The effect of butyrate in bacterial adherence was examined after the co-culture of bacteria with butyrate-treated LoVo cells. Adherence of *L. acidophilus* to butyrate-treated LoVo cells and its inhibitory effect on *E. coli* adhesions are shown in Fig 1A. Treatment with butyrate at 2, 4, or 6 mM resulted in significantly ($p < 0.01$) increased adherence of *L. acidophilus* ATCC 4356 when compared to the control or 8 mM butyrate treatment. Moreover, increased adherence of *L. acidophilus* ATCC 4356 by treatment with butyrate at concentrations of 2, 4, or 6 mM significantly ($p < 0.05$) reduced the adherence of *E. coli* ATCC 43896, suggesting a mutual competition at the binding sites on LoVo cells.

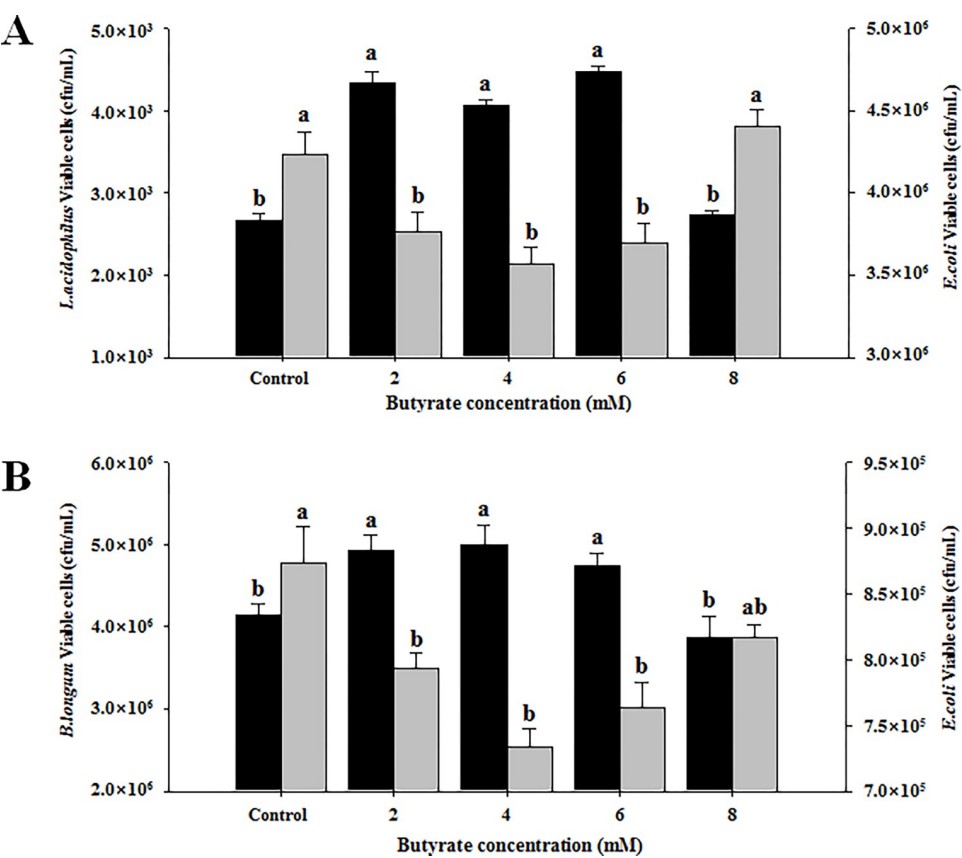

**Fig 1. Adherence of *Lactobacillus acidophilus* ATCC 4356 and *Bifidobacterium longum* ATCC 15707 to LoVo cells and their inhibitory effect on *Escherichia coli* ATCC 43896.** (A) *L. acidophilus* ATCC 4356. Black bar: *L. acidophilus*; grey bar: *E. coli*. (B) *B. longum* ATCC 15707. Black bar: *B. longum*; grey bar: *E. coli*. Values are expressed as mean ± SEM. [abc] Groups with different letters in same row are significantly different by one way ANOVA-test with Duncan's multiple range test ($p < 0.05$).

A similar effect was observed for *B. longum* ATCC 15707. Treatment with butyrate at 2, 4, or 6 mM resulted in a significant ($p < 0.01$) increased adherence of *B. longum* ATCC 15707 in comparison to the control or 8 mM butyrate treatment, which led to significantly ($p < 0.01$) decreased adherence of *E. coli* ATCC 43896 in LoVo cells when compared to the control (Fig 1B). Taken together, these results were consistent with the beneficial effects of butyrate in the homeostasis of the gut microbiome.

## Mucin content and immunohistochemical staining

The result of measuring the released mucin2 content from the butyrate-treated LoVo cells is shown in Fig 2A. Mucin2 content was significantly ($p < 0.01$) increased in LoVo cells treated with butyrate at 4 or 6 mM compared to 0 (control), 2, or 8 mM butyrate treatment. Consistent with the ELISA-based quantification of mucin2, immunohistochemical analysis also showed an increase of the staining intensity in LoVo cells after treatment with butyrate at 4 or 6 mM compared to butyrate treatment at 0, 2, or 8 mM (Fig 2B). The result indicates that enhanced mucin secretion by butyrate treatment would be one of the major effectors in the successful adhesion of the gut microbiota.

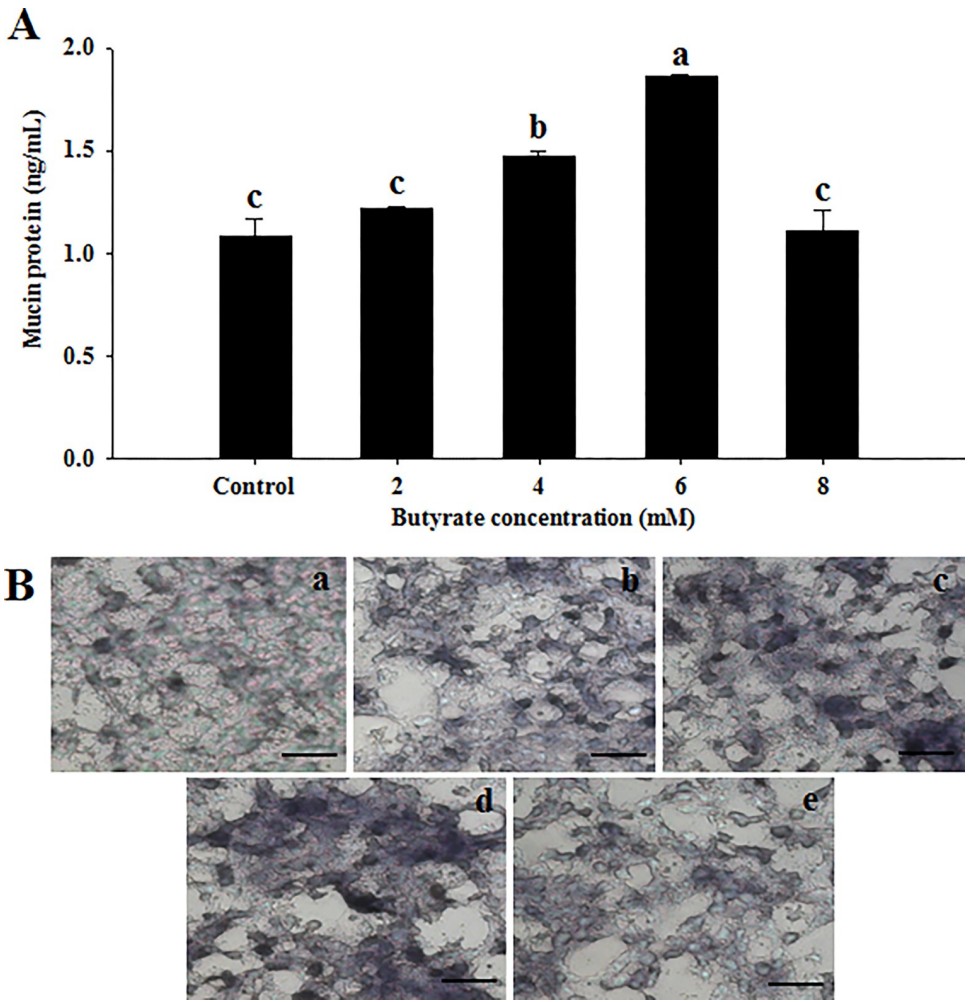

**Fig 2. Effect of butyrate treatment on mucin protein production in LoVo cells.** (A) Mucin protein content. Values are expressed as mean ± SEM. [abc] Groups with different letters in same row are significantly different by one way ANOVA-test with Duncan's multiple range test ($p < 0.05$). (B) Immunohistochemical staining of LoVo cells treated with butyrate at final concentration of 0 (a), 2 (b), 4 (c), 6 (d), and 8 mM (e). Scale bar, 100 μm (×200).

## Mucin secretion pathway of the butyrate-treated LoVo cells

To determine the underlying molecular mechanism of enhanced mucin secretion by butyrate treatment, a TaqMan array 96-well fast plate was utilized to identify the affected signaling pathways. As shown in Fig 3, components of MAPK signaling pathways appear to be upregulated, starting from the Grb2 adaptor protein that links cell membrane-bound receptors to Ras family GTPase proteins. In particular, the Cdc42-PAK signaling pathway that controls microtubules and the actin cytoskeleton appears to be upregulated by butyrate treatment. Other upregulated pathways include the canonical MAPK pathways of ERK, JNK, and p38 MAPKs. Upregulated gene expression in MAPK signaling components and transcription factors of Rel-A (NF-κB) and Fos (AP-1) suggest that butyrate induced pleiotropic effects including cellular proliferation, inflammation, and structural changes in the cytoskeleton.

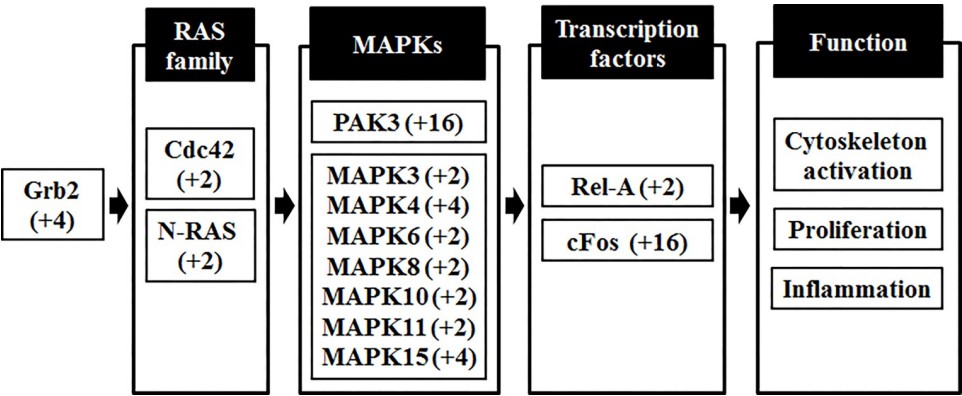

**Fig 3. Analysis of protein kinase signaling pathways in butyrate-treated LoVo cells.** The cDNA converted from total mRNA was applied to a TaqMan array 96-well fast plate, and comparative Ct method was used to calculate relative gene expression levels. The delta Ct value was determined by subtracting the delta Ct of the control sample from the individual Ct of the butyrate-treated samples. Significant changes in butyrate-induced gene expression were screened by filtering genes more than 2-fold increase of transcript compared to the control group. A fold change of the gene expression is shown in the parenthesis. MAPK, mitogen-activated protein kinase.

### Proteomics analysis of the butyrate-treated LoVo cells

Changes in mRNA levels are not always translated to alterations in protein levels due to post-transcriptional or translational control. To complement the limitation of PCR-based transcript analysis, proteomic analysis was conducted to identify differential protein expression directly from LoVo cells. Protein samples from control and butyrate-treated cells were separated by two-dimensional gel electrophoresis using isoelectric focusing and SDS-PAGE and stained with Coomassie G250. Proteins that were highly visible in the butyrate-treated samples but not detectable in the control were isolated and subjected to mass spectrometric analysis (data not shown). Ten protein spots from the butyrate-treated samples were identified by mass spectrometry, as shown in Table 1. The identified proteins belong to categories of metabolic enzymes, protein maintenance, or cytoskeleton that appear unrelated to mucin gene expression. However, preferential extraction of actin and tubulin proteins in LoVo cells would be due to the altered cytoskeletal organization by butyrate treatment. Given the butyrate-induced upregulation of Cdc42-PAK signaling pathway in Fig 3, the enhanced mucin secretion would be partly correlated with actin expression and cytoskeleton polymerization.

**Table 1. Proteins induced by butyrate treatment in LoVo cells.**

| Spot No. | Fold increase | NCBI (gi no.) | Protein Name | Functional Category |
|---|---|---|---|---|
| 1 | +780 | 4557032 | L-lactate dehydrogenase | Energy metabolism |
| 2 | +645 | 32189394 | ATP synthase beta | |
| 3 | +119 | 49457530 | Creatin kinase B | |
| 4 | +1735 | 895845 | P64 CCLP (chloride intracellular channel a) | Signaling |
| 5 | +3255 | 494066 | Glutathione S-transferase (Chain A) | Protein modification and turnover Chaperone |
| 6 | +1703 | 306890 | HSP60 | |
| 7 | +710 | 340021 | Alpha-tubulin | Cytoskeleton |
| 8 | +553 | 4501887 | Actin, cytoplasmic 2 | |
| 9 | +132 | 178045 | Gamma-actin | |
| 10 | +233 | 4501885 | Actin, cytoplasmic 1 | |

Fold increase was calculated by dividing the image density of a spot on the 2D-gel of the butyrate-treated sample by the corresponding value obtained from the control.

## Discussion

Probiotics are reported to improve gut health by maintaining a balanced micro-ecosystem and aiding the host immune response [18]. They can potentially play a protective role against pathogenic species by competing for host cell-binding sites and thus preventing pathogenic bacterial adhesion [19]. Beneficial bacteria including *Lactobacillus* and *Bifidobacterium* strains are known to adhere to surface receptors of enterocytes and inhibit the growth of various pathogenic bacteria, for example, *E. coli* strains that cause food-borne infection, resulting in diarrhea and abdominal pain [20, 21]. It has also been reported that mucin may be an important factor for the removal of pathogens and protection of commensal bacteria in the GI tract [13]. Beneficial bacteria, such as *Lactobacillus* spp., can attach to epithelial cells in the GI tract and induce mucin secretion [22]. Therefore, an increase in mucin secretion may inhibit the adherence of pathogenic bacteria to epithelial cells in the GI tract. Here, an increase in mucin secretion appears to have a different effect in the adhesion of *Lactobacillus*, *Bifidobacterium*, and *E. coli* strains by modulating the binding sites of epithelial cells in the GI tract. Although the underlying molecular mechanism of increased mucin secretion by butyrate treatment remains to be determined the results of this present study indicate that butyrate may stimulate release of mucin in the LoVo cells. Moreover, Gaudier et al. [23] was reported that SCFAs can induce mucin secretion from epithelial cells in the GI-tract, and butyrate appears to be the most effective in promoting mucin secretion. Hatayama et al. [24] reported that butyrate can elevate the mRNA expression of mucin in cell lines derived from human colorectal cancers. These studies showed that butyrate stimulates the secretion of mucin in the colon cell lines. However, the mechanism by which intestinal epithelial cells increase mucin secretion by butyrate treatment needs to be further studied.

In this study, ELISA-based mucin quantification and immunohistochemical staining demonstrated that mucin secretion increased in a dose-dependent manner by butyrate treatment in LoVo cells. However, the results of LoVo cells treated with 8 mM butyrate were not statistically different from those of the control. These results are probably due to the interference of LoVo cell growth caused by butyrate treatment above 8 mM. MUC13 expression has been shown to increase cellular migration and induce F-actin remodeling by PAK1 [25]. This study suggests that MUC13 plays a role in activating the Cdc42-PAK pathway, but not vice versa. Our study demonstrates that butyrate treatment in LoVo cells stimulates both the mucin and MAPK signaling pathways. It remains to be investigated whether stimulated MAPK signaling pathways are a prerequisite for increased mucin secretion in LoVo cells. Inflammatory cytokines have been shown to induce MUC5AC overexpression in human airway epithelial cells through the MAPK signaling pathway [26]. More importantly, AP-1 and NF-B transcription factors containing Fos and Rel-A, respectively, have been identified as key transcription factors that upregulate mucin gene transcription [27, 28]. Therefore, our results are consistent with those of previous studies showing that mucin secretion is controlled by upstream MAPK signaling pathways and transcription factors, AP-1 and NF-B.

Proteomic analysis of LoVo cells showed a notable increase in actin and tubulin levels in butyrate-treated cells. How cytoskeleton dynamics contribute to mucin secretion remains to be investigated. The polymerization of actin microfilaments has been shown to increase mucin secretion under mechanophysical stimulation [29]. It is tempting to speculate that butyrate stimulates MAPK signaling pathways, especially the Cdc42-PAK pathway, to regulate cytoskeleton dynamics, which then increases cellular motility and mucin secretion in target cells [30].

## Conclusion

In the present study, butyrate-treated LoVo cells increased mucin secretion, which stimulated adherence of beneficial bacteria to epithelial cells in the GI tract, with simultaneous inhibition

of the adhesion of pathogenic bacteria. The underlying molecular mechanism of increased mucin secretion by butyrate treatment remains to be determined; however, the MAPK signaling pathways appear to play an important role in the stimulation of mucin secretion in the GI tract. The results of the present study will provide insight into how butyrate modulates bacterial adherence and the MAPK signaling pathway for the homeostasis of colonic epithelial cells.

## Acknowledgments

We would like to thank Editage for editing this manuscript for English language.

## Author Contributions

**Conceptualization:** Kyoung-Sik Han.

**Data curation:** Tae-Hwan Jung.

**Formal analysis:** Tae-Hwan Jung, Jeong-Hyeon Park.

**Investigation:** Tae-Hwan Jung, Jeong-Hyeon Park.

**Methodology:** Kyoung-Sik Han.

**Project administration:** Hyo-Jeong Hwang.

**Supervision:** Hyo-Jeong Hwang.

**Writing – original draft:** Tae-Hwan Jung, Jeong-Hyeon Park.

**Writing – review & editing:** Kyoung-Sik Han, Hyo-Jeong Hwang.

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
