## [Decision Letter · Decision Letter 0]

15 Feb 2022

PONE-D-21-40151Butyrate modulates mucin secretion and bacterial adherence in LoVo cells via MAPK signalingPLOS ONE

Dear Dr. Hwang,

Thank you for submitting your manuscript to PLOS ONE. After careful consideration, we feel that it has merit but does not fully meet PLOS ONE’s publication criteria as it currently stands. Therefore, we invite you to submit a revised version of the manuscript that addresses the points raised during the review process. In particular, a reviewer recommends more information on experimental procedures.

We look forward to receiving your revised manuscript.

Kind regards,

David M. Ojcius

Academic Editor

PLOS ONE

Journal Requirements:

Reviewers' comments:

Reviewer's Responses to Questions

**Comments to the Author**

1. Is the manuscript technically sound, and do the data support the conclusions?

Reviewer #1: Partly

2. Has the statistical analysis been performed appropriately and rigorously? 

Reviewer #1: Yes

3. Have the authors made all data underlying the findings in their manuscript fully available?

Reviewer #1: Yes

4. Is the manuscript presented in an intelligible fashion and written in standard English?

Reviewer #1: Yes

5. Review Comments to the Author

Reviewer #1: 1. Page 6, Line 115-125, ELISA kit to measure mucin section: what type of mucin does this kit target? What type of antibody is used in the kit? What is Reagent A and Reagent B?

2. Page6, Line 132, please specify “primary antibody”. Line 133, what type of secondary antibody used here? Please specify.

3. The authors did not clearly specify the mucin they measured is mucins stored in the cells or the mucins that were released (secreted) outside the cell. In another word, is butyrate able to increase mucin synthesis or secretion, or synthesis and secretion? The protocols provided in the manuscript seems not clear about this.

4. Does butyrate change the mucin types synthesize by the cells? Many types of mucins are produced by GI epithelial cells, they have different functions and properties.

6. PLOS authors have the option to publish the peer review history of their article (what does this mean?). If published, this will include your full peer review and any attached files.

Reviewer #1: No

---

## [Author Response · Author response to Decision Letter 0]

25 Feb 2022

Thank you for your review.

Our researchers did their best to make modifications based on the reviewers' responses.

1. Page 6, Line 115-125, ELISA kit to measure mucin section: what type of mucin does this kit target? What type of antibody is used in the kit? What is Reagent A and Reagent B?

The contents of the ELISA kit were added to the manuscript (page 6, line 116-124)

2. Page6, Line 132, please specify “primary antibody”. Line 133, what type of secondary antibody used here? Please specify.

- Type of the antibodies was mentioned in the manuscript (page 6, line 134-135).

3. The authors did not clearly specify the mucin they measured is mucins stored in the cells or the mucins that were released (secreted) outside the cell. In another word, is butyrate able to increase mucin synthesis or secretion, or synthesis and secretion? The protocols provided in the manuscript seems not clear about this.

- We were measured to the mucins stored in the cells (page 6, line 114)

4. Does butyrate change the mucin types synthesize by the cells? Many types of mucins are produced by GI epithelial cells, they have different functions and properties.

- There were reports that butyrate stimulates the synthesis or secretion of mucin in the colon cell lines. However, the mechanism by which intestinal epithelial cells increase mucin synthesis or secretion by butyrate treatment needs to be further studied. (page 13, 265-267).

---

## [Decision Letter · Decision Letter 1]

11 Mar 2022

PONE-D-21-40151R1Butyrate modulates mucin secretion and bacterial adherence in LoVo cells via MAPK signalingPLOS ONE

Dear Dr. Hwang,

Thank you for submitting your revised manuscript to PLOS ONE. After careful consideration, we feel that it still does not fully meet PLOS ONE’s publication criteria as it currently stands. Therefore, we invite you to submit a newly revised version of the manuscript that addresses the concerns of the reviewer.

We look forward to receiving your revised manuscript.

Kind regards,

David M. Ojcius

Academic Editor

PLOS ONE

Reviewers' comments:

Reviewer's Responses to Questions

**Comments to the Author**

1. If the authors have adequately addressed your comments raised in a previous round of review and you feel that this manuscript is now acceptable for publication, you may indicate that here to bypass the “Comments to the Author” section, enter your conflict of interest statement in the “Confidential to Editor” section, and submit your "Accept" recommendation.

Reviewer #1: (No Response)

2. Is the manuscript technically sound, and do the data support the conclusions?

Reviewer #1: No

3. Has the statistical analysis been performed appropriately and rigorously? 

Reviewer #1: Yes

4. Have the authors made all data underlying the findings in their manuscript fully available?

Reviewer #1: Yes

5. Is the manuscript presented in an intelligible fashion and written in standard English?

Reviewer #1: Yes

6. Review Comments to the Author

Reviewer #1: The authors did not respond or address the questions raised in the previous review. What type of mucin was measured in this study, MUC 1, 2, MUC5AC or MUC6? The issue of mixing mucin synthesis and release (secretion) is not resolved in the revision.

7. PLOS authors have the option to publish the peer review history of their article (what does this mean?). If published, this will include your full peer review and any attached files.

Reviewer #1: No

---

## [Author Response · Author response to Decision Letter 1]

12 Apr 2022

Response to reviewers

Thank you for your review.

Our researchers did their best to make modifications based on the reviewers' responses.

1. Page 6, Line 115-125, ELISA kit to measure mucin section: what type of mucin does this kit target? What type of antibody is used in the kit? What is Reagent A and Reagent B?

The contents of the ELISA kit were added to the manuscript (page 6, line 115-125)

2. Page6, Line 132, please specify “primary antibody”. Line 133, what type of secondary antibody used here? Please specify.

- Type of the antibodies was mentioned in the manuscript (page 6, line 133-134).

3. The authors did not clearly specify the mucin they measured is mucins stored in the cells or the mucins that were released (secreted) outside the cell. In another word, is butyrate able to increase mucin synthesis or secretion, or synthesis and secretion? The protocols provided in the manuscript seems not clear about this.

- We were measured to the measure mucin 2 secreted by the cells. (page 5-6, line 113-114)

4. Does butyrate change the mucin types synthesize by the cells? Many types of mucins are produced by GI epithelial cells, they have different functions and properties.

- There were reports that butyrate stimulates the synthesis or secretion of mucin in the colon cell lines. However, the mechanism by which intestinal epithelial cells increase mucin synthesis or secretion by butyrate treatment needs to be further studied. (page 13, 265-267). 

5. The authors did not respond or address the questions raised in the previous review. What type of mucin was measured in this study, MUC 1, 2, MUC5AC or MUC6? The issue of mixing mucin synthesis and release (secretion) is not resolved in the revision.

- The type of mucin measured in this study is MUC2 secreted by the LoVo cell (page 5-6, line 113-114).

- Although the underlying molecular mechanism of increased mucin secretion by butyrate treatment remains to be determined, the results in the present study indicate that the butyrate treatment may increase mucin secretion in the LoVo cell.

---

## [Editor Report · Decision Letter 2]

14 Apr 2022

PONE-D-21-40151R2Butyrate modulates mucin secretion and bacterial adherence in LoVo cells via MAPK signalingPLOS ONE

Dear Dr. Hwang,

Thank you for submitting your revised manuscript to PLOS ONE. After careful consideration, we feel that it still does not fully meet PLOS ONE’s publication criteria as it currently stands. Therefore, we invite you to submit a revised version of the manuscript that addresses the points raised during the review process. In particular, the reviewer considers that previous concerns were not addressed.

We look forward to receiving your revised manuscript.

Kind regards,

David M. Ojcius

Academic Editor

PLOS ONE

Additional Editor Comments:

The authors did not respond or address the questions raised in the previous review. What type of mucin was measured in this study, MUC 1, 2, MUC5AC or MUC6? The issue of mixing mucin synthesis and release (secretion) is not resolved in the revision.

The authors did not provide any information I requested, such as what is the antibody target? MUC2, MUC5AC, MUC6, MUC7, MUC8, MUC9, MUC20. The issue about mixing up the synthesis and secretion (release) is not resolved in the revised manuscript.

---

## [Author Response · Author response to Decision Letter 2]

26 May 2022

Response to reviewers

Dear reviewers and editorial staffs in PLOS ONE

We are sincerely grateful for your thorough consideration and scrutiny of our manuscript, “Butyrate modulates mucin secretion and bacterial adherence in LoVo cells via MAPK signaling”, manuscript number PONE-D-21-40151R2. Through the accurate comments made by the reviewer, we better understand the critical issues in this paper. We have revised the manuscript according to the Reviewer’s suggestions. We hope that our revised manuscript will be considered and accepted for publication in the PLOS ONE. We acknowledge that the scientific and clinical quality of our manuscript was improved by the scrutinizing efforts of the reviewers and editors. Our researchers did their best to make modifications based on the reviewers' responses and point-by-point responses to the reviewer’ comments are provided below.

Reviewer

1) Reviewer’s comment: (Page 6, Line 115-125) ELISA kit to measure mucin section: what type of mucin does this kit target? What type of antibody is used in the kit? What is Reagent A and Reagent B?

Author’s response: The ELISA kit used in the study targets mucin2. The antibodies used in the kit are capture antibody and biotin-conjugated detection antibody specific to mucin2. The protocol of the ELISA kit included information of reagent A and B were revised in the manuscript (page 6, line 115-130).

2) Reviewer’s comment: (Page6, Line 132) please specify “primary antibody”. Line 133, what type of secondary antibody used here? Please specify.

Author’s response: Primary antibody and a biotinylated secondary andibody used specific to mucin2 (page 6, line 136-137).

3) Reviewer’s comment: The authors did not clearly specify the mucin they measured is mucins stored in the cells or the mucins that were released (secreted) outside the cell. In another word, is butyrate able to increase mucin synthesis or secretion, or synthesis and secretion? The protocols provided in the manuscript seems not clear about this.

Author’s response: We measured the mucin2 that secreted outside the LoVo cells to investigate stimulated by the butyrate treatment in the LoVo cells. We found that in this study, butyrate addition can stimulate the secretion of mucin in LoVo cell, and the protocol has been modified more clearly (page 6, line 115-121).

4) Reviewer’s comment: Does butyrate change the mucin types synthesize by the cells? Many types of mucins are produced by GI epithelial cells, they have different functions and properties.

Author’s response: We think that butyrate may increase mucin produced by cells. However, I would appreciate your understanding that this present study has a limitation of not investigating all types of mucins that can increase in the cell by the butyrate treatment.

Although the underlying molecular mechanism of increased mucin secretion by butyrate treatment remains to be determined, the results of this present study indicate that butyrate may stimulate secretion of mucin in the LoVo cells. Other papers have been also reported that butyrate stimulates the secretion of mucin in the colon cell lines (page 13, line 265-273). 

5) Reviewer’s comment: The authors did not provide any information I requested, such as what is the antibody target? MUC2, MUC5AC, MUC6, MUC7, MUC8, MUC9, MUC20. The issue about mixing up the synthesis and secretion (release) is not resolved in the revised manuscript.

Author’s response: In this study, ELISA kit and immunohistochemical staining kit were used to measure MUC2, and the antibodies also used target for MUC2. More detailed protocols have been modified in the manuscript (page 6, line 115-124; page 6-7, line 136-138) 

The purpose of this study is to investigate the effect of butyrate on mucin released from Lovo cells, and we focused on measuring mucin released from butyrate-treated LoVo cells. We expression for synthesis was deleted and the protocol was more clearly modified so that the secretion and synthesis were not confused in the manuscript (page 6, line 115-121; page 10, line 210-213; page 271-273)

I would appreciate your understanding if there is anything insufficient about the author's response and please give us more detailed comments.

---

## [Decision Letter · Decision Letter 3]

30 May 2022

Butyrate modulates mucin secretion and bacterial adherence in LoVo cells via MAPK signaling

PONE-D-21-40151R3

Dear Dr. Hwang,

We’re pleased to inform you that your manuscript has been judged scientifically suitable for publication and will be formally accepted for publication once it meets all outstanding technical requirements.

Kind regards,

David M. Ojcius

Academic Editor

PLOS ONE

Additional Editor Comments (optional):

Reviewers' comments:

Reviewer's Responses to Questions

**Comments to the Author**

1. If the authors have adequately addressed your comments raised in a previous round of review and you feel that this manuscript is now acceptable for publication, you may indicate that here to bypass the “Comments to the Author” section, enter your conflict of interest statement in the “Confidential to Editor” section, and submit your "Accept" recommendation.

Reviewer #1: All comments have been addressed

2. Is the manuscript technically sound, and do the data support the conclusions?

Reviewer #1: Yes

3. Has the statistical analysis been performed appropriately and rigorously? 

Reviewer #1: Yes

4. Have the authors made all data underlying the findings in their manuscript fully available?

Reviewer #1: Yes

5. Is the manuscript presented in an intelligible fashion and written in standard English?

Reviewer #1: Yes

6. Review Comments to the Author

Reviewer #1: (No Response)

7. PLOS authors have the option to publish the peer review history of their article (what does this mean?). If published, this will include your full peer review and any attached files.

Reviewer #1: No

---

## [Editor Report · Acceptance letter]

6 Jul 2022

PONE-D-21-40151R3 

Butyrate modulates mucin secretion and bacterial adherence in LoVo cells via MAPK signaling 

Dear Dr. Hwang:

I'm pleased to inform you that your manuscript has been deemed suitable for publication in PLOS ONE. Congratulations! Your manuscript is now with our production department. 

Kind regards, 

on behalf of

Dr. David M. Ojcius 

Academic Editor

PLOS ONE